# Endothelial Dysfunction in SARS-CoV-2 Infection

**DOI:** 10.3390/biomedicines10030654

**Published:** 2022-03-11

**Authors:** Francesco Nappi, Sanjeet Singh Avtaar Singh

**Affiliations:** 1Department of Cardiac Surgery, Centre Cardiologique du Nord, 93200 Saint-Denis, France; 2Department of Cardiothoracic Surgery, Aberdeen Royal Infirmary, Aberdeen AB25 2ZN, UK; sanjeetsingh@nhs.scot

**Keywords:** endothelial dysfunction, SARS-CoV-2 infection, thrombosis, angiotensin-converting enzyme-2, angiogenesis

## Abstract

One of the hallmarks of the SARS-CoV-2 infection has been the inflammatory process that played a role in its pathogenesis, resulting in mortality within susceptible individuals. This uncontrolled inflammatory process leads to severe systemic symptoms via multiple pathways; however, the role of endothelial dysfunction and thrombosis have not been truly explored. This review aims to highlight the pathogenic mechanisms of these inflammatory triggers leading to thrombogenic complications. There are direct and indirect pathogenic pathways of the infection that are examined in detail. We also describe the case of carotid artery thrombosis in a patient following SARS-CoV-2 infection while reviewing the literature on the role of ACE2, the endothelium, and the different mechanisms by which SARS-CoV-2 may manifest both acutely and chronically. We also highlight differences from the other coronaviruses that have made this infection a pandemic with similarities to the influenza virus.

## 1. Introduction

The clinical therapeutics that have been suggested in the course of SARS-CoV-2 infection have immediately noted the uncontrolled inflammatory process as an important hurdle to overcome. Endothelial dysfunction and thrombosis are consequences of the cataclysmic inflammatory trigger. Here, we describe the case of a carotid artery thrombosis that occurred in a patient with COVID-19 to highlight endothelial dysfunction during SARS CoV-2 infection. We believe that the data presented in this overview could provide a basis for understanding the role of endothelial dysfunction and assist virologists and health providers (family doctors, internists, cardiologists, and intensivists) in the physician–patient discussion about the risks and expectations after the involvement of the endothelium in SARS-CoV-2 infection.

A 47-year-old woman with no comorbidities tested positive for SARS-CoV-2 via an RNA test, and presented with right upper paresis and headache 15 days after the onset of mild respiratory symptoms. The patient was treated with domiciliary oxygen and managed conservatively prior to the event. Computed tomography (CT) and magnetic resonance imaging (MRI) revealed an ischemic stroke at the junction of the perfusion territories of the left anterior cerebral artery (ACA) and the left middle cerebral artery (MCA). Carotid-specific imaging showed a thrombus of the bulb of the left common carotid artery (LCA) extending into the left internal carotid artery, which was responsible for a 75% vessel stenosis without of any underlying atheroma (Figure 1).

Blood tests results showed mild leucopenia, but no elevation of inflammatory markers including a C-reactive protein (CRP) of <3 mg/L and procalcitonin of 0.02 pg/L. No abnormalities of the coagulation profile were noted and antiphospholipid antibodies, anti-cardiolipin, and autoimmunity screening were negative. Sinus rhythm and echocardiography assessed by ECG ruled out causes of cryptogenic stroke. Anticoagulation by low-molecular-weight-heparin and statin treatment was commenced with clinical improvement and total resolution of the thrombotic occlusion at a 10-day follow-up CT-scan.

## 2. The Clinical Problem

Coronaviruses (CoVs) comprise a large family of single-stranded, positive-sense RNA viruses. They have a capacity for rapid mutation and recombination. Coronaviruses can induce respiratory or intestinal infections in humans and animals [1,2,3].

People who develop acute respiratory infections, including influenza, respiratory syncytial virus, and bacterial pneumonia, may experience progression of the disease to cardiovascular disease (CVD) [4,5,6,7,8]. With the emergence of the severe acute respiratory syndrome virus coronavirus 2 (SARS-CoV-2), which causes coronavirus disease 2019 (COVID-19), the pandemic has rapidly highlighted these major cardiovascular pathologies. The development of CVD as a progression of COVID-19 is generally associated with comorbidities, which can increase the incidence and severity of infectious diseases [7,8,9,10,11]. This rapid progression of the infection has brought out two conditions. In the first, it was highlighted that a large percentage of patients who contract COVID-19 have underlying CVD [12,13]. The second condition revealed that the vascular complications due to COVID-19 are sometimes not related to a pre-existing vascular pathology [13,14,15].

The most relevant finding to emerge from COVID-19 severe acute respiratory syndrome is the devastating global public health crisis. The morbidity and mortality associated with COVID-19 are usually attributed to severe acute respiratory distress syndrome (ARDS) resulting in severe lung function impairment and cardiovascular complications including myocardial infarction (MI). Additionally, disability and death in COVID-19 patients can be caused by ischemic stroke and pulmonary embolism (PE) [14,15,16,17,18,19].

Understanding the effects of COVID-19 on the pulmonary and the cardiovascular system is not only fundamental but allows the provision of complete and satisfactory medical assistance to patients with cardiac-related comorbidities. At the same time, the manifestation of neurological clinical signs, due to endothelial dysfunction with or without systemic atherosclerotic lesions, can occur in patients with COVID-19 [20,21,22]. Prompt diagnosis in these patients can limit the number of neglected cases of infection, reduce delayed diagnosis, and avoid misdiagnosis. In addition, rapid diagnosis allows timely medical treatment and prevents further transmission of the SARS-CoV-2 infection [20,21,22].

Since the beginning of the SARS-CoV-2 pandemic, patients with COVID-19 have been characterized by a higher frequency of arterial and venous thrombosis, which has been linked to systemic inflammation, prolonged bed rest, and a prothrombotic environment [23,24]. There is now abundant evidence that arterial and venous thromboembolism (VTE) is a major cardiovascular risk in patients with COVID-19, potentially leading to neurological complications [25,26,27,28]. The percentage of patients experiencing VTE is higher in intensive care unit (ICU) admissions with symptomatic disease reaching 25%, but the rate stands at 69% when routine venous ultrasound scans are performed [27,28]. Several studies have reported a high prevalence of microthrombosis in situ, which may be related to endothelial damage directly caused by viral infection [14,15,16,28,29].

Concerns related to quantifying the risk of cardiovascular and neurological complications depend on the heterogeneity of the patient population with COVID-19 and direct SARS-CoV-2 infection of the endothelium remains unclear [30], with both prohibiting accurate determination of risk. In addition, the different methods of thromboprophylaxis, the definitions of variable outcomes, and the restriction of assessments in the ICU setting should be considered. Although antithrombotic therapy is recommended for patients hospitalized with COVID-19, with the aim of preventing thromboembolic cardiovascular events, arterial and venous thromboembolism still occurred in a subgroup of patients who had received standard thromboprophylaxis at the appropriate time [31,32,33].

## 3. Pathophysiology

### 3.1. Coronavirus Infection in Humans

The name “coronavirus” derives from the presence of the crown-shaped spikes on their surface. Coronaviruses belong to the *Coronavirinae* subfamily, which is further classified into four phylogenetic groups: the CoV α, β, γ, and δ. The α and β groupings are pathologically harmful because they can cause infection in humans [7,34,35]. Coronaviruses consist of four main structural proteins: the peak (S) protein, which mediates the binding of the viral particle to the host receptor and subsequent fusion of the virus and cell membrane. The nucleocapsid (N) protein, the membrane protein (M), and the envelope (E) proteins also belong to the structure of the coronavirus [6,36].

In the mid-1960s, investigators identified the first human CoV (HCoV) from embryonic tracheal organ cultures and, until 2003, only two HCoV species were acknowledged as pathogens: HCoV-229E and HCoV-OC [37]. In 2003, two other types of HCoV emerged, leading to severe acute respiratory syndrome coronavirus (SARS-CoV)40 and Middle East Respiratory Syndrome coronavirus (MERS-CoV) [38]. In 2019, a newly identified SARS-CoV-2 caused one of the more lethal respiratory infections in humans [9,11,39,40,41,42,43,44]. Thus, there are currently seven known human coronaviruses that have the potential of infecting humans but three of these strains are highly pathogenic (SARS-CoV, SARS-CoV-2, and Middle East respiratory syndrome (MERS)). The less virulent CoV strains, including HCoV-229E, HCoV-NL63, HCoV-OC43, and HCoV-HKU1, are often self-terminating infections and are usually termed as “common colds” [42,45].

### 3.2. SARS Cov-2 Host Interaction

We learned that cells are the gateway to viral infection through a binding interaction between viruses and the host’s cell surface and this interplay is mediated by a certain affinity with surface receptors. Viral trophism towards a given cell line is regulated by the expression and distribution of structured receptors on the cell surface that mediate virus entry. Therefore, receptors represent the crucial feature that specifically defines tissue infection as well as the pathogenesis of the disease.

Virologists discovered that SARS-CoV-2 is the third human coronavirus that uses the enzyme angiotensin-converting peptidase 2 (ACE2) as the gate of entry for cells [46]. Understanding the mechanism that regulates the interaction between SARS-CoV-2 and ACE2 is a fundamental step in determining both the trophism of the virus for tissues and the progression of SARS-CoV-2 infection, as well as its evolution towards more severe forms of COVID-19. As a result, the knowledge of the cellular processes that form the cornerstone of SARS-CoV-2 infection could represent the turning point in the identification of treatments that prevent the progression of the infection towards the development of complicated disease, thus favoring a reduction in mortality.

ACE2 is configured as a transmembrane protein that plays the main role in homeostasis of the cardiovascular system in counterbalancing the effects of the angiotensin-converting enzyme (ACE) [47]. ACE acts as a catalyst in the conversion of angiotensin I to angiotensin II. This octapeptide is very active in exerting a vasopressor action, mediating vascular contraction with an increase in blood pressure, and in promoting a proinflammatory activity. ACE2 is a carboxypeptidase that is active in the conversion of angiotensin II into the heptapeptide angiotensin-(1-7) to facilitate its function as an antagonist of angiotensin II. This work is mainly expressed by exercising anti-inflammatory and vasodilatory activity. Substantial evidence suggests elevated expression of ACE is in vascular endothelial cells of the lungs, and it is therefore likely that the level of angiotensin II is also increased in lung vascular cells. In support of this finding, there is evidence that in mouse models where acute lung injury was induced, ACE2 deletion causes more severe disease, thus suggesting a protective action of ACE2 in lung tissue due to its role in reducing the levels of the pro-inflammatory angiotensin II [47].

Since ACE2 is the SARS-CoV-2 receptor and as reported in many cases of virus–host interaction, viral receptor expression is down-regulated in infected cells. This genetic regulation with reduced expression of the ACE2 receptor was noted in the lungs of mice with SARS-CoV infection. Lung damage caused by SARS-CoV and SARS-CoV-2 can be caused by the depletion of ACE II, which therefore plays a central role in maintaining the infection, as also evidenced by the high angiotensin II levels that were reported in the plasma of patients with COVID-19. However, patients with MERS-CoV infection experienced lung disease similar to that induced by SARS-CoV-2 without the direct involvement of ACE2. Therefore, other factors are implicated in the genesis of coronavirus-mediated infection (Figure 2).

### 3.3. Functional Characteristics of Glycoprotein S

As with all coronaviruses, SARS-CoV-2 also has a 180 kDa (S) spike as a structured protein, which serves to identify the key to access cells in the ACE2 receptor. Protein S plays two essential roles: it induces binding to ACE2 from the amino-terminal region, and it promotes the fusion of viral and cell membranes through the carboxy-terminal region [48]. The third crucial step for lung cell infection is the proteolytic activation of spikes that occurs at a host-mediated polybasic furin cleavage site [49,50]. Evidence has shown that this cleavage site characterizes all spike proteins from patients who have clinically manifested SARS-CoV-2 infection. It is important to note that the furin cleavage site is typical of other highly pathogenic viruses such as the influenza A virus, but is not expressed by SARS-CoV.

This scientific finding has prompted researchers to consider that its acquisition likely occurs after recombination with coronaviruses in bats. Therefore, in the light of this discovery, it is possible to state that cellular tropism of SARS-CoV-2 is multiplied by the cleavage process sustained by the furin proteases and could represent the turning point in explaining how transmission from bats to humans has been promoted [51]. However, the tropism of SARS-CoV-2 depends on expression of other cellular proteases that act synergistically to ACE2. Indeed, additional proteases are required for cleavage and act in the fusion of SARS CoV-2 with the host-cell membrane.

In particular, the transmembrane serine protease 2 (TMPRSS2) and TMPRSS13 are host-cell-surface proteases that cleave the spike shortly after binding to ACE2 [49]. Following the interaction of protein S with the ACE2 receptor, host-cell-surface proteases such as TMPRSS2 and TMPRSS13 cleave the full-length spike protein (S0), converting it to its S2 site through a complex mechanism mediated by the selective function of the host’s furin. Activation of S glycoproteins of SARS CoV-2 (SARS-2-S) by these surface proteases requires processing of the S1/S2 cleavage loop, in which both the furin recognition motif and extended loop length have proven to be critical [52,53].

Virology taught us that the greater transmissibility of SARS-CoV-2 is related to relevant replication in the upper airways, which is not disclosed for the other highly pathogenic coronaviruses SARS-CoV and MERS-CoV. The S glycoproteins of different coronaviruses revealed an intrinsic temperature preference, corresponding with the temperature of the upper or lower airways. SARS-2-S and HCoV-229E, which are responsible for the common cold, replicate in an optimal way when the temperature of 33 °C is produced as in upper airways. Instead, the S proteins of SARS-CoV and MERS-CoV work better at 37 °C, in accordance with the favorable virus-replication preference for the lower airways. TMPRSS2 and TMPRSS13 proteases amplify the cell tropism of SARS-CoV-2 because they facilitate SARS-2-S-driven access into the host cell through its activation. Both proteases were found to be relevant in the context of authentic virus replication albeit with differences. For example, TMPRSS13 is not active in low-pathogenic HCoV-229E because it is not effective as a spike activity amplifier [52].

The process of cleavage at the S2 site facilitates the fusion of viral and cell membranes to deliver viral RNA into the cytosol [54,55]. In some defined conditions that have been reported, cells exhibit low TMPRSS2 expression, which allows the possibility of undertaking alternative pathways of virus uptake, including those involving the endolysosomal pathway and cathepsins [52]. Evidence suggests that the D614G mutation increases the stability of SARS-CoV-2, particularly at 37 °C, and improves its utilization of the cathepsin L alternative pathway. The use of a subsidiary route for virus entry may confer greater stability of the S-glycoprotein. In these two years, we have learned that the spike’s properties can promote the spread of the virus and potentially explain why the spike-G614 variant that replaced the first D614 variant has become globally predominant. Coronavirus spike protein is adjusted to suit airway temperature and protease conditions to improve virus transmission and pathology [52].

Neuropilin 1 (NRP1) has recently emerged as a protein that works in the fusion between the cell and SARS-CoV-2 for its entry [56,57,58]. Neuropilin has the role of a receptor that binds the RXXR motif carboxyterminal sequence of the spike exposed after furin cleavage. The precise mechanism of NRP1 in promoting the entry of SARS CoV-2 into cells remains to be clearly determined, but its role may be to amplify infection by involving other types of cells such as endothelial cells [53,59] (Figure 3).

## 4. Endothelial Cell Infection and Endotheliitis in SARS-CoV-2 Infection

SARS-CoV-2 infection has very heterogeneous characteristics causing several clinical syndromes of COVID-19 when an involvement of the vascular endothelium is established. The two extreme clinical conditions experienced by patients with COVID-19 are asymptomatic infection or a fatal disease. Evidence from critical illness hospitalizations due to COVID-19 disclosed about 30% of patients in which a serious disease with progressive lung damage occurred [54,60]. The unfortunate progression of this pathology is associated with the severe injuries of the vascular compartment resulting in rupture of the vascular barrier and edema. It should be noted that in the patho-anatomical examination of the lesions, endotheliitis, thrombosis, and marked infiltration of inflammatory cells were associated [61].

Several studies have suggested that vascular pathology is a major cause of severe disease. As proof of this, investigators have identified thrombotic and microvascular complications as the potent cause of deaths in patients with complicated COVID-19 [28,62,63]. Arterial and venous thromboembolism, kidney disease, and neurological disorders are mandatory among the pathological events responsible for the onset of severe symptoms in COVID-19 patients requiring hospitalization [64,65] (Figure 4).

This suggests a leading role of SARS-CoV-2 in activating the vascular system in a consistent process, which is potentially responsible for the multi-organ involvement of the infection and the consequential multi-organ failure. However, solid evidence confirming whether viral infection leads to multi-organ failure or that the latter is caused by inflammation-induced endothelial activation remains highly controversial and is still an open topic.

As for the cardiovascular system, evidence of its serious involvement leading to complications from SARS-CoV-2 infection was apparent very soon after the respiratory disease. It has been unquestionably highlighted that these complications pose a key threat in COVID-19. With the advent of COVID-19 we have learned that pre-existing CVD represents a signifi-cant risk factor in patients with SARS-CoV-2 infection who develop COVID-19. The mecha-nisms that sustain the disproportionate rate of cardiovascular complications in patients who experienced SARS-CoV-2 infection remain unclear. However, once SARS-CoV-2 in-fects the myocardium, it can cause direct or indirect damage. Likewise, in these patients outcomes are worse than in patients who do not exhibit CVD [14,15,16].

### 4.1. Interaction of SARS-CoV-2 and ACE2 Receptor: Insight of Influence on Renin–Angiotensin–Aldosterone System Inhibitors

The tissue and circulating elements that constitute the renin–angiotensin system (RAAS) establish an elaborate intersecting network of regulatory and counter-regulatory peptides. As previously reported, ACE2 has the key counter-regulatory function mediated by the degradation of angiotensin II to angiotensin-(1-7). ACE2 has the effect of containing the vasoconstriction, sodium retention, and fibrosis induced by the production of angiotensin II. Although the primary substrate of ACE2 is angiotensin II, the action of ACE2 is also the catalysis of angiotensin I into angiotensin-(1-9) and it interferes with the hydrolysis of other peptides [66]. ACE2 was found to be expressed in human tissue specimen from 15 organs but it was also found to be widely expressed in the heart and in the kidneys, as well as on the pulmonary alveolar epithelial cells, which are the target cells of SARS-CoV-2 infection [67]. It is important to note that the circulating levels of soluble ACE2 are scarcely understood, as is the functional role that ACE2 plays in the lungs under normal conditions [68]. This underactive role carried out by ACE2 changes towards an upregulation in some clinical states.

Evidence has shown that ACE inhibitors and angiotensin receptor blockers (ARBs) have different effects on angiotensin II, which is the primary substrate of ACE2, so these agents can be expected to exert different activity on ACE2 levels. Although substantial structural homology has been demonstrated between ACE and ACE2, their enzymatic active sites are different. As a result, ACE inhibitors in clinical use do not exhibit an action to unequivocally affect the activity of ACE2 [69]. Diverging results regarding the effects of ACE inhibitors on ACE2 levels have been reported in experimental animal models [70,71,72,73]. By evaluating results from the animal models, inconsistent evidence on the effects of ARBs on ACE2 are disclosed. In fact, some of these experiences have reported the effect of ARBs in increasing messenger RNA expression or ACE2 protein levels in tissues [70,74,75,76], while in other studies, no consequence about its role was proven [77]. On the other hand, a relevant number of human studies investigating the effects of RAAS inhibition on ACE2 expression have not yielded positive results. Campbell et al., reported no difference in angiotensin-(1-7) production after intravenous administration of ACE inhibitors in patients with coronary artery disease. The observation that stems from this discovery is the lack of substantial evidence that ACE inhibitors have direct effects on the ACE2-directed metabolism of angiotensin II [78]. Luque et al., obtained comparable results by analyzing angiotensin-(1-7) levels in patients who experienced hypertension and reported a lack of effect after initial treatment with the selective ACE inhibitor captopril. However, patients who were undergoing six-to-six continuous monotherapy with captopril revealed an increased level of angiotensin-(1-7) [79]. Furuhashi et al., retrospectively analyzed a longitudinal cohort of Japanese patients with hypertension who reported increased urinary ACE2 levels among patients who were managed with long-term ARB olmesartan, compared with the group of control patients who received no medical treatment. However, this association has not been disclosed with the use of the ACE inhibitor enalapril or with other ARBs such as losartan, candesartan, valsartan, and telmisartan [80].

The observation that emerges in the evaluation of these conflicting results suggests that a complex mechanism exists at the basis of the RAAS responses to the pathway modulators. Another point that reinforces the lack of definitive evidence from human studies, thus lifting the veil of uncertainty, concerns the results provided by preclinical models that may not be readily noted in physiological conditions. This point is clearly emphasized in the study of Furuhashi et al., in which the effects on ACE2 after administration of RAAS inhibitors should not be interpreted as uniformly applicable data because the response to therapies within a given drug class was also different [80].

A precise observation concerns the plasma level of ACE2, which may not be a reliable indicator of the activity of the integral structured form in the membrane. In fact, ACE2 is released from the membrane by a process that appears to be supported by a separate regulation mediated by an endogenous inhibitor [81]. Another important observation concerns the degree of expression and biological relevance of ACE2, which is not homogeneous as it may vary according to the type of tissue and the clinical condition. Unfortunately, our knowledge sustained by solid data supporting the effects of ACE inhibitors, ARBs, and other RAAS inhibitors on the lung-specific expression of ACE2 in animal and human experimental models is inconsistent. Furthermore, even assuming that RAAS inhibitors can modify the levels or activity of ACE2 in the target microcirculation, there is no substantial clinical data proving the involvement of RAAS inhibitors in facilitating the interaction of the spike protein of SARS-CoV-2 with ACE2. Further studies on ACE2 receptor mechanics need to be undertaken in humans as they would serve to better define the distinctive interaction between SARS-CoV-2 and the RAAS network.

### 4.2. RAAS Blockers in SARS-CoV-2 Infection—Potential for Benefit or Harm?

The biomolecular mechanisms that regulate the interaction between SARS-CoV-2 and ACE2 lead not only to the initial entry of the virus into the host cell via ACE2 but also induce a subsequent downregulation. The crucial effect of downregulation concerns the lack of protective effect on organs exerted by ACE2. No hypothesis has ever demonstrated that the continuous activity of angiotensin II may be partly responsible for organ damage in patients with COVID-19 [82,83]. SARS-CoV-2 saturating the ACE2 involved in the binding with the spike leads to the subsequent down-regulation of the fullness of ACE2 receptors in host cells’ surface [47]. In the early stage of SARS-CoV-2 infection, viral infection and replication work together to decrease membrane ACE2 expression. The initial infiltration of neutrophils was subsequently observed in response to bacterial endotoxin, which can be induced by the down-regulation of ACE2 activity in the lungs [84]. Under these conditions, an unconstrained accumulation of angiotensin II with local RAAS activation occurs. Liu et al., studied a small series of patients with COVID-19 who reported high plasma levels of angiotensin, which were consequently related to a higher total viral load and the degree of lung damage [83] (Figure 5).

A dysregulation of ACE2 can theoretically also lead to decreased cardioprotection in the context of myocardial involvement following pulmonary and circulatory haemodynamic compromise in patients who develop COVID-19 with severe critical conditions [85,86]. In severe forms of COVID-19 with a course characterized by grave clinical deterioration preceding death, high levels of markers of myocardial injury were revealed [87]. These markers underwent a rapid increase in parallel with worsening clinical conditions [60]. After all, the cardiotropicity of many viruses has been reported, confirming that very high viremia associated with a wide range of infectious agents can be responsible for subclinical viral myocarditis.

ACE2 is involved, with a well-recognized role, in myocardial recovery and injury response, as reported in the study by Oudit et al., in which autopsy findings of patients who died from SARS were examined. In 35% of the heart samples studied, the authors recorded the presence of viral RNA, which was related to a reduced expression of the ACE2 protein [88]. In support of this study, it was reported that in human explanted hearts with dilated cardiomyopathy, the administration of recombinant ACE2 leads to a normalization of angiotensin II levels [89].

On the basis of this evidence, numerous randomized studies have been designed, the aims of which were to verify whether the administration of a recombinant ACE2 protein can be useful in restoring the balance of the RAAS network and, therefore, is potentially able to prevent organ damage (ClinicalTrials.gov number, NCT04287686). In addition, two ongoing RCTs have been designed based on the administration of losartan in the treatment of COVID-19 in a non-hospitalized patient group (NCT04311177) and to patients requiring hospitalization (NCT04312009). Patients enrolled in these RCTs had not received prior medical treatment with the administration of a RAAS inhibitor.

### 4.3. Insights of Angiogenesis and ACE 2 Expression on Endothelial Cells in SARS-CoV-2 Infection

The topic is controversial with a large number of studies that reported evidence of ACE2 expression on endothelial cells [28,38,90,91,92]. ACE 2 is expressed in several organs, including the lung, heart, kidney, and intestine, but endothelial cells also have ACE2 on their surface.

In vitro experiments have reported that SARS-CoV-2 can directly infect organoids in engineered human blood vessels [93]. However, this evidence does not prove that the vascular disorders in COVID-19 are due to the involvement of the endothelial cells by the virus.

Ackerman et al., reported that vascular angiogenesis, either intussusceptive or germinative, differentiated the pulmonary pathobiology of patients who had COVID-19 from that of patients with similar severe influenza virus-related infections. Given this background, investigators examined seven lungs of patients who died due to COVID-19 that were compared with those who died from acute respiratory distress syndrome (ARDS) secondary to influenza A (H1N1). These autopsy specimens were separated to 10 age-matched, uninfected control lungs.

Several points of Ackerman’s study deserve detailed assessments. First, immunohistochemical analysis data concerning ACE expression in controls of patients without lung infection suggested ACE2 expression in alveolar epithelial cells (0.053 ± 0.03) and capillary endothelial cells (0.066 ± 0.03). In contrast, in the lung findings of patients with COVID-19 and those with influenza A who had died as a result of the onset of severe respiratory failure, the relative ACE2-positive tallies revealed a high expression of alveolar epithelial cells (relative counts of 0.25 ± 0.14 vs. 0.35 ± 0.15) and endothelial cells (relative counts of 0.49 ± 0.28 and 0.55 ± 0.11), respectively. It is important to note that ACE2-positive lymphocytes were not disclosed in perivascular tissue or alveoli of the non-infected controlled lungs. However, the interaction between the ACE2 receptor and immune cells occurred in the lungs of patients with COVID-19 and in those who had experienced H1N1 infection in combination with a respiratory distress syndrome (relative counts of 0.22 ± 0.18 and 0.15 ± 0.09, respectively) [28].

The second major finding reported was the occurrence of a marked angiogenesis process that intervenes in patients who died as a result of severe COVID-19. This population had consistently demonstrated intussusceptive angiogenic lung features that were significantly higher (60.7 ± 11.8) compared to those of the lungs of patients who had suffered from H1N1 infection (22.5 ± 6.9) or to those of patients without any signs of lung infection (2.1 ± 0.6). The comparison was statistically significant for both populations (*p* < 0.001). With regard to the consistency of conventional germinative angiogenesis features, it was also greater in the COVID-19 population compared to the influenza population [28].

The third interesting finding of this study involved a correlation between pulmonary angiogenesis and hospitalization, where angiogenesis was plotted as a function of the length of hospital stay. Investigators noted that the degree of intussusceptive angiogenesis was found to be significatively affected with the raising of the duration of hospitalization (*p* < 0.001). In contrast, examination of the lungs of patients with H1N1 infection revealed less intussusceptive angiogenesis without a demonstrable increase over time. A similar pattern was seen for germinative angiogenesis [28].

Finally, Ackermann et al., provided an evaluation of angiogenesis-related genes that was performed in patients in which associated respiratory failure occurred. Investigators, using a multiplex analysis of angiogenesis-related gene expression, examined 323 genes from the nCounter PanCancer progression panel (NanoString Technologies) and recorded differences between samples from patients with COVID-19 and those from patients with influenza. A total of 69 angiogenesis-related genes were found to be differentially regulated in deceased COVID-19 patients, compared with 26 differentially regulated genes that were disclosed in patients who experienced the H1N1 infection. The relevant finding was that 45 genes shared changes in expression [28].

A potential greater ACE2 receptor expression caused by induced angiogenesis can lead to a substantially higher interplay of SARS-CoV-2 with the receptor so as to implement ACE2′s power in sustaining the infection. However, other investigations have assumed the primary role in the involvement of microvascular pericytes, as these types of cells express high levels of the ACE2 receptor [90,94,95]. Therefore, it is the lesion of the pericyte, from the extension of the viral infection, that is the determining cause inducing endothelial dysfunction.

In particular, Mc Cracken et al., investigated ACE2 expression in human endothelial cells (ECs) and the ability of SARS-CoV-2 to infect the endothelium by analyzing transcriptomic and epigenomic data on ECs. Viral proteins combined with ECs were studied in vitro using an analysis of RNA sequencing. The ENCODE has allowed the genetic evaluation of infected ECs of different tissues including arterial, venous, and microvascular beds, in comparison with epithelial cells from respiratory, gastrointestinal, and skin sources. Investigators recorded a very low or no basal ACE2 expression in ECs compared with epithelial cells. Moreover, they observed that in vitro exposure of ECs to inflammatory cytokines was increased in the plasma of patients with severe COVID-19 [95]. This condition failed to upregulate ACE2 expression. Single-cell RNA sequencing of human organ donor arts revealed that, while ACE2 sequence reads were abundant in pericytes, they were rare in ECs [96]. Although there was an unusual endothelial ACE2 expression, contamination from adherent pericyte fragments made it possible to determine a common confounder in vascular single-cell RNA sequencing data [97]. Evidence suggested that there was a scarce occurrence of ACE2 transcripts in human heart ECs and this was likely determined by pericyte contamination. Investigators exposed the ECs to SARS-CoV-2 and observed that replication levels were extremely low, even after the ECs were exposed to very high concentrations of the virus compared to more permissive VeroE6 cells. The observed low levels of SARS-CoV-2 replication in ECs were due to viral entry via an alternative pathway to the ACE2-dependent one. This non-ACE-receptor-mediated entry was attributable to exposure of greater concentrations of the virus. The reported data proved that direct endothelial infection by SARS-CoV-2 is not likely to occur. The endothelial damage disclosed in patients with severe COVID-19 and in critically ill conditions was more likely secondary to infection of neighboring cells. Other mechanisms, including immune cells, platelets, and complement activation, and circulating pro-inflammatory cytokines were implicated in endothelial damage. This evidence was supported by current achievements proving that plasma from critically ill and convalescent patients with COVID-19 leads to EC cytotoxicity [98].

### 4.4. Other Molecules (Mediators) Affecting Angiogenesis during SARS-CoV-2 Infection

Reports have demonstrated the expression of neuropilin receptors [53,56,57,58,59] and TMPRSS2 [99], thus suggesting a viral infection of endothelial cells.

Investigators have learned that neuropilin 1, a type-1 membrane protein, was characterized by being highly expressed among Xenopus frogs, chicken, and mice. The extracellular component of NRP-1 was composed of three distinctive domains, each of which was involved in molecular and/or cellular interactions. Evidence suggested that in mammals, NRP-1 was expressed in the cardiovascular system, the nervous system, and in the limbs at specific phases of development. Studies performed on chimeric embryos revealed several morphological abnormalities such as excess capillaries and blood vessels, dilation of blood vessels, malformed heart as well as ectopic sprouting and defasciculation of nerve fibers. These kinds of abnormalities experienced in chimeric embryos suggested that NRP-1 has several functions in embryonic morphogenesis [56,57,58].

Gerhardt et al., disclosed a role of NRP-1 in central nervous system germinative angiogenesis that was found to be mediated by binding to the isoform VEGF165 (Vascular Endothelial Growth Factor). The latter led to the stimulation of angiogenesis and was indispensable for the development of cerebral vessels in the mouse.

Targeted inactivation of the NRP1 receptor can lead to abnormalities in the blood flow that are capable of influencing the dynamics of SARS-CoV-2 infection. Therefore, blood fluid dynamics plays a crucial role in the effects of SARS-CoV-2 infection. The vascular neoformation and the presence of flow changes can induce both stabilization of the infection in the primary localization and an acceleration in the propagation of the virus, making it potentially the determining factor in the dynamics of SARS-CoV-2 infection. Jones et al., revealed that the loss of NRP1 function rather than the induced dynamic flow change was the cause of the altered capillary plexus geometry. The evidence was in favor of vessel remodeling defects that occurred concomitantly with the onset of blood flow. A genetic alteration in neuropillin1 mutants led mainly to a deficiency related to endothelial cell migration rather than replication. The inadequacy of migration of endothelial cells was, therefore, the main cause of the altered blood flow dynamics [57].

Aspalter et al., worked on VEGF and Dll4/Notch signaling, which cooperate in a negative feedback circle. VEGF and Dll4/Notch signaling play a central role in the specified endothelial tip and stalk cells to oversee the function of the vessels. Thus, VEGF and Dll4/Notch were active in sprouting angiogenesis by guiding the growth of blood vessels in healthy and diseased tissues. Investigators discovered the key endothelial function of NRP1, which abolished the stalk-cell phenotype by restricting the activation of Smad2/3 through the function of Alk1 and Alk5. The evidence first revealed that Notch downregulates Nrp1, thereby relieving the inhibition of Alk1 and Alk5, thus driving stalk-cell behavior. Second, the authors disclosed that the heterogeneity between neighboring endothelial cells determined by the lateral Dll4/Notch feedback circle used NRP1 levels as the pivot, which in turn warranted a distinctive reactivity to TGF -β/BMP signaling [58].

More recently, two reports focused on the specific role of NRP1 as the key factor implicated in the viral trophism. Canuti-Castelveteri et al., evaluated the role of nuropillin1 in the tissue tropism because NRP1 works similarly to a cofactor on the surface of the host cell in promoting easier interaction between the viral particle and the virus receptor. The authors suggested that NRP1 significantly reinforces SARS-CoV-2 infectivity by working to bind furin-cleaved substrates. This action was prevented with the use of a monoclonal blocking antibody against NRP1. In the case of mutations induced in SARS-CoV-2, undergoing a modification in the furin cleavage site, the infectivity of SARS-CoV-2 was not potentiated by neuropillin1. Human autopsies performed on the olfactory epithelium after COVID-19 provided pathological findings in which SARS-CoV-2-infected NRP1-positive cells were present in the nasal cavity. The reported data added important information on the infectivity of SARS-CoV-2 cells in the first stage and defined a potential target for the development of antiviral drugs [59].

Daily et al., found that the host furin protease recognizes an attachment site on the viral protein S that cleaves the full-length precursor of S glycoprotein into two associated polypeptides: S1 and S2. This interaction facilitating the cleavage of protein S led to the creation of a polybasic Arg-Arg-Ala-Arg carboxy-terminal sequence on S1. This sequence conformed to a C-end rule (CendR) motif, which has the characteristic of selectively binding NRP1 and NRP2 to cell-surface receptors. The relevant data that emerged was the selectivity of the CendR motif in S1, which directly linked NRP1. Once the blocking of this interaction had been carried out using RNA interference or through selective inhibitors, a reduced entry of SARS-CoV-2 could be obtained, which resulted in reduced infectivity. Therefore, NRP1 was shown to act as a host factor for SARS-CoV-2 infection and could potentially provide a selective therapeutic target to counter COVID-19 [53] (Figure 6).

Hofmann et al., proved that the serine protease TMPRSS2 was employed after the engagement of the ACE2 receptor by SARS-CoV-2 and that it worked via S protein priming, and they disclosed that SARS-CoV-2′s spread also depends on TMPRSS2 activity. The role of TMPRSS2 was pivotal and synergistic with furin-mediated pre-cleavage at the S1/S2 site in infected cells, which could ultimately encourage subsequent SARS-CoV-2-dependent TMPRSS2 entry into target cells. This mechanism can be compromised with a TMPRSS2 inhibitor camostat mesylate which is approved for clinical use in Japan. This substance has potentially an increased antiviral activity leading to a blockage of viral entry and it might constitute an off-label treatment option [45].

## 5. The Pathoanatomic Alteration of the Endothelium and SARS-CoV-2 Infection

Investigations on autopsy findings of patients who died after COVID-19 have helped to clarify whether SARS-CoV-2 can directly infect the endothelium. Bryce at al reported the first 100 COVID-19-positive autopsies performed at the Mount Sinai Hospital in New York City noticing the presence of large pulmonary emboli in six cases. Furthermore, the authors found that diffuse alveolar damage occurred in over 90% of cases and that microthrombi were discovered as typical lesions in multiple organs including the brain. Hemophagocytosis was another peculiar patho-anatomic lesion. Electron microscopic tests revealed the presence of the virus in the samples, and laboratory results from the COVID-19 cohort revealed high levels of inflammatory markers, abnormal clotting values, and elevated cytokines IL-6, IL-8, and TNFα [100].

In the autopsy checks by Ackerman et al., an interesting finding was a shared histological picture in the peripheral lung in both patients who died from respiratory diseases associated with COVID-19 and those who had influenza-associated respiratory failure. Both showed diffuse alveolar damage with perivascular T cell infiltration. However, the patients who had suffered from COVID-19 disclosed well-defined and characteristic pulmonary vascular changes. Consistent histopathological evidence suggested the occurrence of a severe endothelial lesion associated with the presence of intracellular viruses and destroyed cell membranes. An analysis of the pulmonary vessels of patients with COVID-19 revealed widespread thrombosis with microangiopathy. Damage generated by alveolar capillary microthrombi was nine times more prevalent in patients with COVID-19 and patients with influenza, compared to healthy lung findings (*p* < 0.001). The data that differentiated the findings of patients who died from COVID-19 to those for influenza-associated respiratory failure concerned the greatest amount of growth of new vessels in the lungs of patients with COVID-19. This conspicuous microvascular growth was mainly attributable to a mechanism of intussusceptive angiogenesis and was 2.7 times as high as that of the lungs of patients with influenza (*p* < 0.001) [28].

Histopathology revealed, in all lung specimens from the COVID-19 autopsies, a spread alveolar injury that may be disclosed as focal with only mild interstitial edema or occurred with homogeneous fibrin accumulation associated with marked interstitial edema and early intraalveolar organization. Necrosis of alveolar lining cells, pneumocyte type-2 hyperplasia, and linear intraalveolar fibrin deposition were observed in either focal or diffuse lesions. The difference was noted in specimens of the influenza group in which florid diffuse alveolar damage with massive interstitial edema and extensive fibrin deposition occurred in all cases. In addition, in some cases, specimens presented focal organizing and resorptive inflammation. These changes were suggestive of the substantially higher weight of the lungs from patients who died from influenza [28].

A cytological evaluation revealed differences between the autopsy findings of patients who died from influenza and those who died from COVID-19. In the lungs from patients with COVID-19 and patients with influenza CD3-positive T cells (26.2 ± 13.1 for COVID-19 and 14.8 ± 10.8 for influenza), these values referred to 200-μm radius of precapillary and postcapillary vessel walls in 20 fields of examination per patient. With the same field size, the authors found a more relevant amount of CD4-positive T cells in lungs from patients with COVID-19 than in lungs from patients with influenza (13.6 ± 6.0 vs. 5.8 ± 2.5, *p* = 0.04), whereas the amount of CD8-positive T cells was also significant (5.3 ± 4.3 vs. 11.6 ± 4.9, *p* = 0.008). The number of neutrophils (CD15 positive) was significantly lower adjacent to the alveolar epithelial lining in the COVID-19 group than in the influenza group (0.4 ± 0.5 vs. 4.8 ± 5.2, *p* = 0.002) [28].

Schaefer et al., studied the in situ expression of SARS-CoV-2 in airways and lungs obtained at autopsy of seven autopsy cases (male, N = 5; female, N = 2) with confirmed COVID-19 infection. The use of reverse transcriptase-polymerase chain reaction (RT-PCR)-validated SARS-CoV-2 infection and the detection of viral particles from autopsy cases were evaluated using a rabbit polyclonal antibody against SARS Nucleocapsid protein in correlation with clinical parameters. Chest imaging suggested a widespread-airspace disease occurred in all patients. Histologic examination revealed an acute diffuse alveolar damage (DAD) in five cases, while two cases developed preferentially organized alveolar injuries. Among the five patients with acute diffuse alveolar damage, SARS-CoV-2 was located in pulmonary pneumocytes, while in all cases, damage involved ciliated airway cells. In two cases, viral particles were identified also in the upper airway epithelium. Interestingly, in two patients with organizing DAD, SARS CoV-2 was not discovered in lungs or airways, and in these cases, the investigators did not reveal endothelial cell infection. This evidence strongly suggested that SARS-CoV-2 infection, involving epithelial cells in the lungs and airways of patients with COVID-19 who progressed toward respiratory failure, could be identified during the acute stage of lung injury and was absent in the organizing stage [101].

Varga et al., found the pathological modifications of the endothelium exposed to SARS-CoV-2 infection in patients of different ages and with multiple comorbid conditions. The authors described the case of a recipient of a renal transplant with coronary artery disease and arterial hypertension. In the patient, the deterioration of clinical conditions due to COVID-19 required the use of mechanical ventilation. On the eighth day, after being admitted to the ICU, the evolution towards a severe form of multisystem organ failure had resulted in death. The use of electron microscopy performed on post-mortem autopsy findings highlighted the presence of viral inclusion structures in endothelial cells in the transplanted kidney. The histological analysis showed the accumulation of inflammatory cells associated with the endothelium. The histological changes involved many tissues, including apoptotic bodies, in the heart, the small bowel, and lung. The lung experienced a remarkable accumulation of mononuclear cells with characteristic involvement of most small lung vessels that appeared congested [102].

Severe endotheliitis occurred in patients with comorbidities such as diabetes, arterial hypertension, and obesity. A severe and progressive respiratory failure due to negative evolution of COVID-19 led these patients to hospitalization in the ICU. Subsequently, the impairment of the clinical condition may lead to multi-organ failure. Mesenteric ischaemia is generally caused by thrombosis and endotheliitis, requiring prompted removal of necrotic small intestine. From a cardiocirculatory point of view, severe right heart failure may occur as an evolutionary progression of left ventricular compromise related to consequent ST-segment elevation and myocardial infarction. The progressive evolution of this complication was toward a cardiac arrest. Histological post-mortem analysis recorded a high grade of lymphocytic endotheliitis in lung, heart, kidney, and liver as well as a rapid progression towards necrosis of liver cells. In addition, pathoanatomic damage with evidence of myocardial infarction was reported, but without any proof of pathoanatomic injuries related to lymphocytic myocarditis. The small intestine was investigated and the histology revealed endotheliitis of the submucosal vessels [102].

Delorey et al., studied donors who died of COVID-19 producing both single-cell atlases from 24 lung, 16 kidney, 16 liver, and 19 heart autopsy tissue samples and spatial atlases of 14 lung samples. With the use of integrated computational analysis, investigators found the occurrence of substantial remodelling in the lung epithelial, stromal, and immune sections. In these three compartments, evidence suggested multiple tracks of failed tissue regeneration, including defective alveolar type-2 differentiation and expansion of fibroblasts and putative TP63+ intrapulmonary basal-like progenitor cells [103]. The authors enriched viral RNAs in phagocytic mononuclear and endothelial lung cells, which induced host-specific programs of response to infection. For example, investigators differentiated lung regions with and without viral RNA expression so that spatial analysis in the lung distinguished host inflammatory responses. Instead, through the non-spatial analysis performed on other tissue atlases, transcriptional alterations were recorded in multiple cell types in the heart tissue of donors with COVID-19. Based on genome-wide association studies of COVID-19, a mapping of the cell types and genes involved was performed and correlated with disease severity. This study highlighted fundamental data that clarify the biological effect of severe SARS-CoV-2 infection by highlighting the systemic character of the infection involving the whole body, suggesting a fundamental step towards new treatments [103].

Lindner et al., revealed the existence of SARS-CoV-2 in the myocardial tissue from autopsies of patients who died. The objective of investigation was to reveal a possible cardiac response to the infection [104] by working at discovering the incidence of SARS-CoV-2 positivity in cardiac tissue by employing selective immuno-investigation of CD3+, CD45+, and CD68+ cells in the myocardium. In addition, they evaluated the gene expression of tumor necrosis growth factor α, interferon γ, chemokine ligand 5, as well as interleukin-6, -8, and -18 in myocardial tissue of patients infected with SARS CoV-2. The study included 39 consecutive autopsy cases with a median (interquartile range) patient age of 85 (78–89) years, with 59.0% being women. The presence of SARS-CoV-2 was recorded at a rate of 61.5% of patients and viral load with a concentration of 1000 copies per μg of RNA was reported at a rate of 41.0%, respectively. As regards cytokine response, the panel disclosed the existence of six proinflammatory genes that were increased in 16 patients who experienced SARS-CoV-2 in the heart compared to the 15 patients without any localization of infection. The comparison between 15 patients without infection located in the heart and 16 patients who recorded more than 1000 copies revealed the absence of inflammatory cell infiltrates or differences in leukocyte numbers per high-power field. The results of this study performed on autopsy cases could suggest the presence of SARS-CoV-2 within the myocardium. Although it might speculate a potential response to SARS-CoV-2 infection that could be produced in cases with higher virus load vs. no virus infection, there is no clear evidence that the presence of the virus in the myocardium leads to an influx of inflammatory cells. Future investigations should be directly focused on evaluating the long-term consequences of cardiac involvement and the influx of inflammatory cells [104] Table 1.

## 6. The Role of the Endothelium in Infection: Direct or Vicarious?

Thrombosis and microangiopathy are relevant in lung tissue infected with SARS-CoV-2. An analysis of the pulmonary vasculature of the lungs in the COVID-19 patients who succumbed to respiratory complications revealed the prevalence of thrombi in pulmonary arteries with a diameter of 1 mm to 2 mm at the level of precapillary vessels, without complete luminal obstruction. Similar pathoanatomic alterations were common in the autopsy surveys of patients infected with the influenza A virus. Nonetheless, a distinctive feature between the two pathogens located in pulmonary vasculature was the formation of thrombi in the capillary alveoli. Although they were disclosed in both biopsy findings, fibrin thrombi occurred nine times more commonly in patients with COVID-19 compared to influenza. Contrary to the arteriolar system, the venular one seems more affected by the infection sustained by Virus A with a statistically higher incidence of intravascular thrombi in postcapillary venules of less than 1 mm in diameter. These histologic findings were supported by three-dimensional micro-CT of the pulmonary specimens suggesting that the lungs from patients with COVID-19 and influenza showed nearly total occlusions of precapillary and postcapillary vessels [28].

The unresolved focal question is whether SARS-CoV-2 can replicate in endothelial cells of the new angiogenic process and whether the more or less distorted neo-architecture can be considered a factor influencing the replication.

The findings by Schimmel et al., go against the trend and the researchers suggest that in vivo infection of endothelial cells by SARS-CoV-2 is unlikely. They speculate that endothelial infection can occur in the condition defined as basolateral infection, which occurs only if the adjacent lung epithelium is denuded. Another possibility includes the potential infection of the endothelium when a high viral load is present in the blood, a condition that the investigators define as apical infection. In the scenario described, while the occurrence of SARS-CoV-2 infection of the endothelium is conceivable, it does not contribute to viral amplification [105]. However, the central role of endothelial cells in SARS-CoV-2 infection plays a key role in the pathogenesis of the infection. The progression of infection from the pulmonary epithelium with the involvement of the adjacent infected endothelium favored the development of a pro-inflammatory response to SARS-CoV-2 [105].

An interesting finding was provided by examining the microvascular architecture of the lungs of patients with COVID-19. The autopsies of the lungs of patients with COVID-19 experienced distorted vascularity with structurally deformed capillaries. This chaotic angiogenesis revealed the presence of capillaries that had sudden changes in caliber and the presence of pillar intussusception within the capillaries. Ultrastructural damage of the endothelium was disclosed on electron microscopy examination of the endothelium within which the presence of intracellular SARS-CoV-2 was observed. In these pathoanatomical circumstances, the investigators did not rule out the possibility of identification of the virus in the extracellular space as well [28].

## 7. Comments and Conclusions

Although the studies reporting autopsy findings were scarce in number, the vascular features identified were consistent with the presence of distinctive cardiac, pulmonary vascular, and endothelial pathobiological features in these cases of COVID-19 [28,100,101,102,103,104].

Important evidence for the understanding of endothelial damage and angiogenesis was provided by the study of autopsy lung findings. The lungs from the patients who suffered from COVID 19 revealed morphologic patterns of diffuse alveolar damage and infiltrating perivascular lymphocytes. Reports have disclosed three distinctive angiocentric features of COVID-19. First, evidence has proved severe endothelial injury coupled with intracellular SARS-CoV-2 virus and disrupted endothelial cell membranes. Secondly, robust findings suggest widespread vascular thrombosis with microangiopathy and occlusion of alveolar capillaries that occurred in the lungs of patients with COVID-19. Third, the lungs of patients with COVID-19 remarkably revealed an increase in new vessel growth through a mechanism of intussusceptive angiogenesis [28,106,107]. In particular, intussusceptive angiogenesis in the lungs from patients with COVID-19 was characterized by new vessel growth that might have occurred by conventional sprouting or nonsprouting angiogenesis. It is important to note that the existence of a pillar or pivot crossing the lumen of the vessel was the distinctive feature of intussusceptive angiogenesis [108]. These critical changes in the endothelial-lined intravascular structure, which were specific to the appearance of the intussusceptive pillar, were not visible under the light microscope but they could be easily identified by corrosion casting and scanning electron microscopy [28].

Ackerman et al., showed that in addition to tissue hypoxia, the greater degree of endotheliitis and thrombosis in patients’ lungs who experienced COVID-19 could cause a high rate of germination and intussusception angiogenesis observed in these patients. Although the authors suggested that the degree of intussusceptive angiogenesis in patients with COVID-19 had significantly increased with the increasing length of hospitalization, no definitive conclusions could be drawn about the presence of these lesions and the disastrous clinical evolution [28]. The authors previously reported that intussusceptive angiogenesis was the main angiogenic mechanism directly implicated in the late stages of chronic lung injury [28].

The clinical neurological manifestations during SARS-CoV-2 infection that develop following the involvement of the vascular system responsible for cerebral circulation deserve further attention. Neurological manifestations in the course of COVID-19 infection have been reported as being as high as 36% in a large 226-patient cohort, with five cases of stroke. A pathogenetic link between COVID-19 infection and systemic hypercoagulability or prothrombotic state was suggested [22]. Other cases were associated with anti-cardiolipin and antiphospholipid antibodies [109]. Cases of thrombotic neurological events have been also sporadically reported over the course of this pandemic. Wang et al., reported a series of five COVID-19 patients with cerebral vessel occlusion treated with mechanical thrombectomy. In this series, patients’ investigations demonstrated a large clot burden with fragmentation and the involvement of multiple territories. In all these cases, an underlying disturbance of the coagulation profile was demonstrated [17]. A case of acute ischemic stroke from a large floating thrombus within the common carotid artery in a patient with no specific past medical history or risk factors was also recently described. MRI showed a large thrombus adherent to a thin atheromatous plaque but failed to demonstrate ulceration, hemorrhage, or signs of arteritis. However, a hypercoagulable state with elevated D-dimer and CRP was shown, corroborating the hypothesis of a thrombotic proclivity triggered by the infection-related inflammatory response and explaining the occurrence of thrombi in relatively unusual sites [18]. Interestingly, Fara et al., reported three cases of stroke secondary to large vessel thrombosis without occlusion even in the setting of a mild infection with COVID-19, in which the inflammatory response is supposedly expected to be weaker. However, they still found a significant systemic hypercoagulability with elevated CRP and D-Dimer [19].

In our case, symptomatic large-vessel arterial thrombosis occurred in the absence of a significant alteration of the coagulation profile or thrombophilic state and without a significant laboratory-demonstrable inflammatory response. No atheromatous disease was demonstrable. This case might support the emerging hypothesis of a direct effect of SARS-CoV-2 on plaque stability and endothelium function [110]. Viral particles have been identified within the endothelium with accompanying endotheliitis and apoptosis. Viral-mediated endothelial disruption was considered responsible for endothelial dysfunction [102] and might lead to local thrombus formation even independently of the systemic pro-inflammatory effect of the infection. However, an isolated, direct effect of the pathogen in the context of such a systemic infection is difficult to discern and there is no direct proof of viral-related endothelial damage in this case as we could not perform histological examinations.

Finally, the role of ACE2 is fundamental in mechanisms leading to endothelial dysfunction. ACE2 is an integral membrane protein that appears to be the host-cell receptor for SARS-CoV-2 [49,111]. A significantly higher number of ACE2-positive cells in the autopsy of lungs from patients with COVID-19 has been revealed. Histopathological evidence demonstrated higher numbers of ACE2-positive endothelial cells and crucial modifications in endothelial morphology and this finding provides further confirmation of a steady head role of endothelial cells in the vascular phase of COVID-19. Typical disruption of intercellular junctions, cell swelling, and a loss of contact with the basal membrane were observed in endothelial cells from selected samples of patients with COVID-19. Several studies testified the presence of the SARS-CoV-2 virus within the endothelial cells, suggesting that direct viral effects, as well as perivascular inflammation, may lead to critical contributions to the endothelial injury [28,102,112].

## Figures and Tables

**Figure 1 biomedicines-10-00654-f001:**
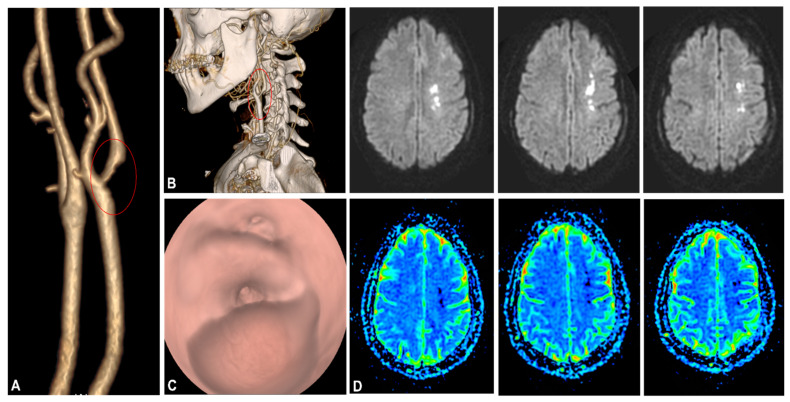
(**A**,**B**): CT angiogram with three-dimensional reconstruction depicting thrombotic stenosis of the left carotid artery bifurcation extending into the internal carotid artery (red circles). (**C**): CT Virtual Intravascular Endoscopy of the left carotid bifurcation demonstrating a large thrombus causing significant stenosis of the left carotid artery. (**D**): MRI brain diffusion-weighted imaging showing punctiform lesions in hypersignal diffusion visible in left fronto-parietal FLAIR at the level of the left ACA-MCA territory, with decreased apparent diffusion coefficient. No hemorrhagic changes. Gyriform enhancement in relation to the lesions indicating a rupture of the blood–brain barrier. Preserved patency of circle of Willis. Abbreviation: ACA, anterior cerebral artery; CT, computed tomography; MCA, middle cerebral artery; MRI, magnetic resonance.

**Figure 2 biomedicines-10-00654-f002:**
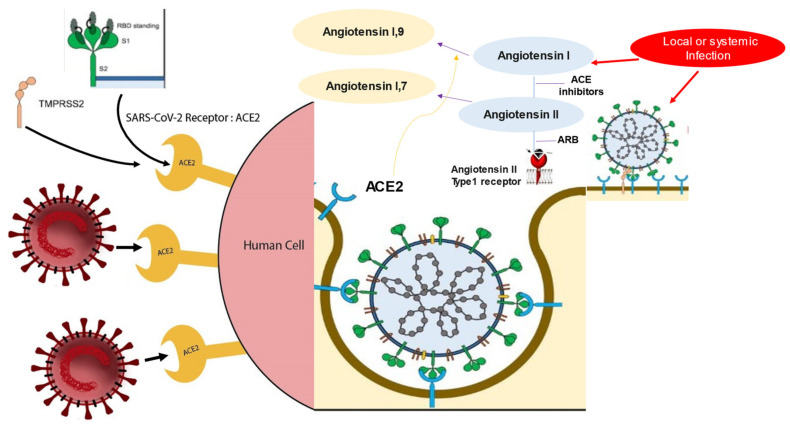
SARS-Cov-2 uses the enzyme angiotensin converting peptidase 2 (ACE2) as the gate of entry for cells. The trophism of the virus for tissues leads to the inflammatory profile pattern before and after SARS-CoV-2 infection. The initial entry of severe acute respiratory syndrome coronavirus 2 (SARS-CoV-2) into cells is shown with involvement mainly of type-II pneumocytes. S glycoproteins of SARS-CoV-2 (SARS-2-S) bind to its functional receptor, the angiotensin-converting enzyme 2 (ACE2). Host-cell-surface proteases such as TMPRSS2 cleave the full-length spike protein (S0), converting it to its S2 site through a complex mechanism mediated by the selective function of the host’s furin. After endocytosis of the viral complex, surface ACE2 is further down-regulated, resulting in obstacle-free storage of angiotensin II. Local activation of the renin–angiotensin–aldosterone system may mediate lung injury responses to viral injuries. Abbreviations: ACE2, angiotensin-converting enzyme 2; ARB, angiotensin-receptor blocker; CVD, cardiovascular disease; RBD, receptor-binding domain; SARS-CoV-2, severe acute respiratory syndrome coronavirus 2; TMPRSS2, transmembrane serine protease 2.

**Figure 3 biomedicines-10-00654-f003:**
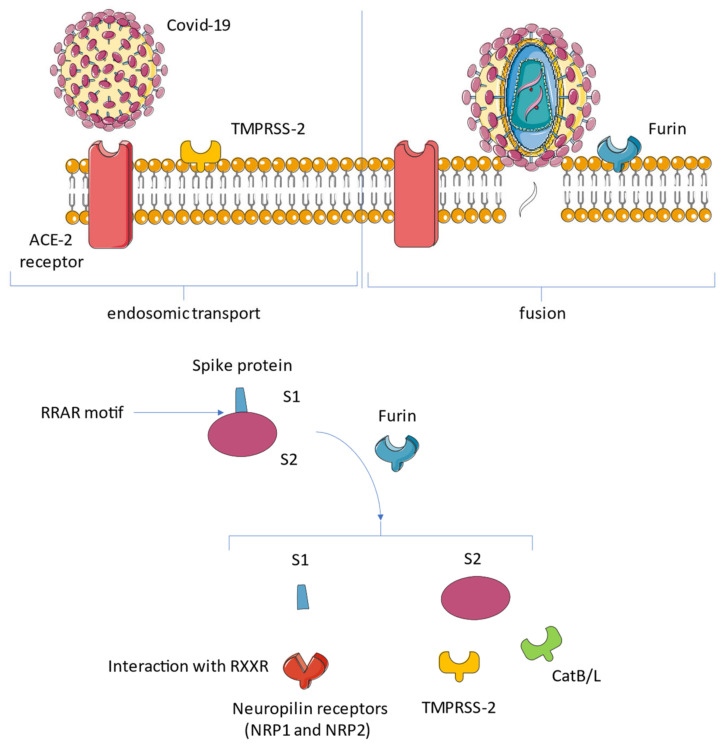
The tropism of SARS-CoV-2 depends on the expression of proteases that act synergistically to ACE2. Transmembrane serine protease 2 (TMPRSS2) and TMPRSS13 cleave the full-length spike protein (S0), converting it to its S2 site through a complex mechanism mediated by the selective function of the host’s furin. Neuropilin 1 (NRP1) binds the RXXR motif carboxyterminal sequence of the spike exposed after furin cleavage. Abbreviations: CatB/L, cathepsin B/L; NRP1, neuropilin; S, spike; other abbreviations are given in the above Figure 3 image.

**Figure 4 biomedicines-10-00654-f004:**
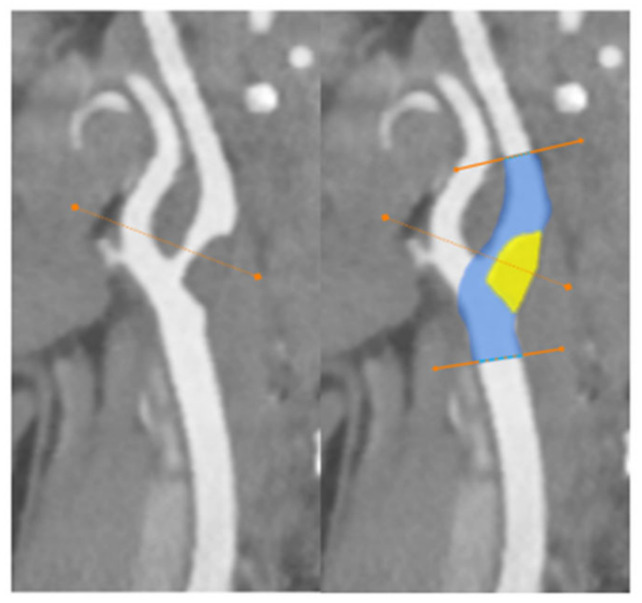
Endothelial cell infection and endotheliitis in critically ill hospitalized patients with COVID-19. CT-angiography of the supra-aortic arteries disclosed a significant stenosis of the left carotid artery bifurcation extending into the internal carotid artery.

**Figure 5 biomedicines-10-00654-f005:**
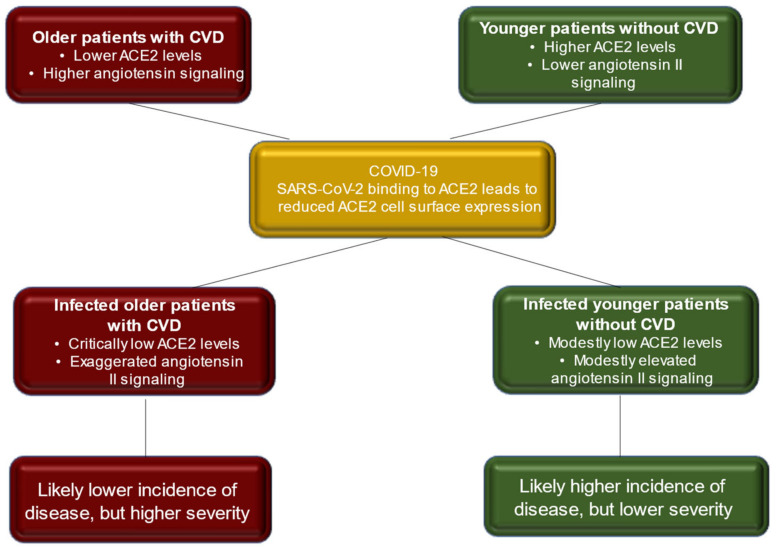
Depiction of the inflammatory profile of COVID-19 patients before and after Coronavirus SARS-CoV-2 infection. The scheme reveals the occurrence of two different degrees of disease in the investigated populations.

**Figure 6 biomedicines-10-00654-f006:**
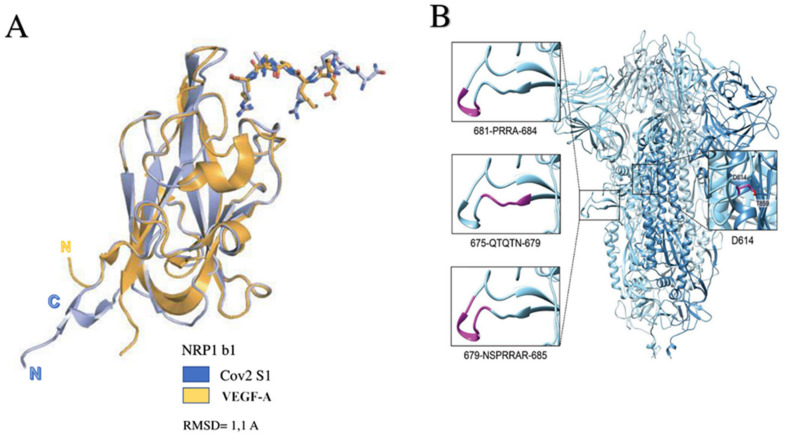
(**A**): Depicts the NRP1 b1–S1 CendR peptide complex superposed with the NRP1 b1–VEGF-A fusion complex (based on PDB ID: 4DEQ †). Bound peptides are disclosed in stick representation. (**B**) Structure of the SARS-CoV-2 spike trimer, based on PDB 6ZGE ††, in which we modelled the cleavage loop using SWISS-MODEL. Transmembrane serine protease 2 (TMPRSS2) and TMPRSS13 cleaves the spike shortly after binding ACE2. Host-cell-surface proteases such as TMPRSS2 and TMPRSS13 cleave the full-length spike protein (S0) converting it to its S2 site through a complex mechanism mediated by the selective function of the host’s furin. Activation of S glycoproteins of SARS-CoV-2 (SARS-2-S) by these surface proteases requires processing of the S1/S2 cleavage loop, in which both the furin recognition motif and extended loop length are critical. Abbreviations: RMSD, root-mean-square deviation; VEGF-A, Vascular Endothelial Growth Factor; other abbreviations are given in previous figures. SARS-CoV-2 Sequence Resources. Genome Reference Sequence (NC_045512) [55,56,57,58,59].

**Table 1 biomedicines-10-00654-t001:** Studies reporting pathoanatomic alteration in SARS-CoV-2 infection. Abbreviations: DAD, diffuse alveolar damage; other abbreviations are given in previous figures.

First Author/Year Ref	Type of Study	Number of Patients	Mean Age (Yrs)	Autopsy (n)	Findings
Bryce et al., 2021 [100]	OS	100	29 to 94 years (Median 68)	Lung 99 Heart 97 Spleen 86 Lymphnodes 60 Kidney 94	82 cases DAD; Hemophagocytosis Higher cytokines IL-6, IL-8, and TNFα.
Ackerman et al., 2020 [28]	OS	14 SARS-CoV-2 7 H1N1 7	68 ± 9.2 years (female) 80 ± 11.5 years (male)	Lung 14	Alveolar capillary microthrombi 9 times more in SARS-CoV-2 Higher CD3, CD4 and CD-8 positive T cells in SARS-CoV-2 Lower neutrophils (CD15)
Schaefer et al., 2020 [101]	OS	7	50 to 77 (Median 66) Male 16 Female 23	Lung 7	5 cases diffuse DAD; 2 cases alveolar injuries. SARS-CoV-2 infection involving epithelial lung cell in acute phase No endothelial cell infection
Varga et al., 2020 [102]	OS	3	58 to 61 years (Median 63)	Kidney 2 Lung 2 Heart 1 Liver 1 Intestin 2	Lymphocytic endotheliitis in lung, heart, kidney, and liver. Apoptotic bodies in the heart Mononuclear cells in lung
Delorey et al., 2021 [103]	OS	32	30 to 89 years Male 20 Female 12	Kidney 16 Lung 24 Heart 19 Liver 16	Higher viral RNAs in phagocytic mononuclear and endothelial lung cells. Transcriptional alterations in multiple cell types in the heart tissue.
Lindner et al., 2020 [104]	Prospective	39	78 to 89 years (Median 68) Male 16 Female 23	Heart 39	SARS-CoV-2 infects directly the myocardium Absence of inflammatory cell infiltrates in patient with SARS-CoV-2 infection. Higher cytokine response

## Data Availability

Not applicable.

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
