# Peer review of "Endothelial Dysfunction in SARS-CoV-2 Infection"

_biomedicines, 2022, doi:10.3390/biomedicines10030654_

Round 1

Reviewer 1 Report

The review by Nappi and Singh is an excellent review of the literature on the topic of SARS CoV2 infection and endothelial damage. Starting with a clinical case, the authors do a comprehensive review of the literature on the evidence in preclinical and clinical settings of the contribution of viral infection on endothelial function. In a systematic and critical manner, the authors discuss the role of the RAA system, ACE2, neuropilin, and TMPRSS, as well as the role of infection-associated inflammation and direct viral infection in endothelial damage demonstrated by autopsy studies.

The review is well written, clear, well-articulated, and the figures, diagrams, and table are sufficient to guide the reader in understanding the text.

To change the graphic abstract with figure 2.

Author Response

The authors thank the reviewer for the comment.
The graphical abstract has been changed to figure 2

Reviewer 2 Report

This is a review article, focused on endothelial dysfunction in COVID-19. This is an important issue, and this field has been reviewed mainly in 2020. This review article was updated, and this reviewer considers that the authors have well written this review paper. This reviewer has some comments as described below.

Major comments:

  1. In pages 1-2, the authors reported their case, which had thrombosis in left carotid artery. In response to this case, the authors went to the next section, “2. The Clinical Problem”, and they stated about arterial and venous thrombosis. However, in page 4, Graphical Figure, the authors mentioned about only “VTE”, but not arterial thrombosis. They should include arterial thrombosis in Graphical abstract.
  2. Throughout the whole manuscript, how did the authors consider the difference between arterial and venous (pulmonary arteries) thrombosis in COVID-19? If there are differences, they should describe in another section regarding the differences with figure(s).
  3. There are several groups of COVID-19; alpha, beta, delta, and omicron. In the clinical situation, there are differences in vascular damages among these groups. The authors should show the differences in this issue.
  4. Page 8, lines 9 from the bottom. What is “edema h”?
  5. Page 17, para. 2. The authors mentioned “figures A-D”, but the paper did not have these figures. They should show them.

Author Response

Response to reviewer

Authors thankful the reviewer for Her/His comment

1. In pages 1-2, the authors reported their case, which had thrombosis in left carotid artery. In response to this case, the authors went to the next section, “2. The Clinical Problem”, and they stated about arterial and venous thrombosis. However, in page 4, Graphical Figure, the authors mentioned about only “VTE”, but not arterial thrombosis. They should include arterial thrombosis in Graphical abstract.

The Graphical abstract was changed with Figure 2   as suggested also  by    reviewer 1

2. Throughout the whole manuscript, how did the authors consider the difference between arterial and venous (pulmonary arteries) thrombosis in COVID-19 ? If there are differences, they should describe in another section regarding the differences with figure(s).

The differences are not related to the inflammatory pathophysiopathological process that determines the endothelial disfunction. They depend on its localization which can be either at the venular level or at the arteriolar level as shown on page 15 and highlighted in yellow

3. There are several groups of COVID-19; alpha, beta, delta, and omicron. In the clinical situation, there are differences in vascular damages among these groups. The authors should show the differences in this issue.

Endothelial dysfunction results from the production of cytokines. There is no evidence that has shown a different characteristics of inflammatory response to the type of coronavirus responsible for the infection. The difference is in the severity of the disease. For example the case presented in the manuscript was a Wuhan 1 variant of the Coronavirus. As for the Omicron variant, the main difference concerns its location in the upper respiratory tract.

4. Page 8, lines 9 from the bottom. What is “edema h” ?

   Errors have been removed

5. Page 17, para. 2. The authors mentioned “figures A-D”, but the paper did not have these figures. They should show them.

    Errors have been removed

Round 2

Reviewer 2 Report

The authors have revised this review paper. This reviewer still has a comment as described below.

Major comment:

Revised Figure 2. In this Figure, the authors mentioned only VTE, not arterial thrombus. Do the authors consider that the prevention of arterial thrombosis is not appropriate in here? In the clinical settings, VTE prophylaxis seems to work as the arterial thrombosis prevention. Is it OK that arterial thrombosis prevention in Figure 

Author Response

Authors thankful the reviewer for Her/His comment

All figures are checked.
In yellow the clarifications suggested by reviewer